# Innovative Freeze-Dried Snacks with Sodium Alginate and Fruit Pomace (Only Apple or Only Chokeberry) Obtained within the Framework of Sustainable Production

**DOI:** 10.3390/molecules27103095

**Published:** 2022-05-11

**Authors:** Agnieszka Ciurzyńska, Paulina Popkowicz, Sabina Galus, Monika Janowicz

**Affiliations:** Department of Food Engineering and Process Management, Warsaw University of Life Sciences (WULS-SGGW), 159c Nowoursynowska St., 02-776 Warsaw, Poland; agnieszka_ciurzynska@sggw.edu.pl (A.C.); s186435@sggw.edu.pl (P.P.); sabina_galus@sggw.edu.pl (S.G.)

**Keywords:** freeze-drying, snack, fruit pomace, hydrocolloids, structure

## Abstract

The aim of the work has been to develop freeze-dried fruit snacks in the form of bars with the use of by-products derived from fruit processing. In effect 14 product designs of fruit gels based on: apple pulp, apple juice, water, sodium alginate and only apple or only chokeberry pomace were prepared. The snacks were freeze-dried. The dry matter content, water activity, structure, colour, mechanical properties, as well as organoleptic evaluation were determined. Freeze-dried bares were obtained according to sustainability production which in this case was relied on application of fruit pomace. The freeze-drying process ensures the microbiological safety of the product without the need to use chemical preservatives. Freeze-dried samples obtained low water activity in the range of 0.099–0.159. The increase in pomace concentration (3–9%) boosted the dry matter content to above 98%, and decreased the brightness of the freeze-dried bars about 6 to 10 units. Mechanical properties varied depending on the product design. With the increase in the amount of pomace, the shear force increased at 23% to 41%. Based on the results, the best variant, that has the most delicate structure and the best organoleptic properties, has proven to contain 1% sodium alginate and 2% pomace.

## 1. Introduction

The sustainable development is a conceptual framework that provides for the quality of life at the level of the current civilisational development. The idea of the sustainable development was created as part of the activities of the United Nations World Commission for Environment and Development, the so-called Brundtland Commission established in 1983. Since then, it has addressed the problems of the contemporary world [1]. The production of food in a traditional way may be inefficient, therefore it is important to use raw materials, means of production that do not have such an adverse impact on the natural environment, society and the national economy. In addition, the sustainable development refers to the lack of dependence on energy sources that may run out over time, and the marketing of products that have a positive impact on human health [2].

The fruit and vegetable industry in Poland in 2018/2019 processed approx. 1.3 million tons of vegetables and 0.97 million tons of fruit [3]. As a result of processing fruit into juice, drink, cider or wine, by-products are obtained to be the so-called pomace that consists of seeds, remains of pulp and skins [4]. Their composition depends mainly on the processing conditions and the fruit variety. After the production process, about 20% of residual waste remains from the raw material. Since the pomace contains a lot of water (often over 80%), it is easily microbiologically contaminated. This makes the pomace a perishable product and entails various storage difficulties, mainly due to limited storage space in processing plants and inadequate preservation of the pomace, which may have a negative impact on the natural environment. The storage problems are even greater due to the structural and chemical diversity and the differences in the intended use of the pomace. Preserved (by drying or ensiling) or pre-treated waste would be suitable for further use [5].

Pomace, especially apple pomace, accounts for the largest percentage share in the fruit and vegetable industry (over 90%) [6,7]. Apple pomace is a by-product of the fruit processing [8]. One of the ways to manage apple pomace is to add dried pomace to dietary food [8,9,10]. In the course of chokeberry fruits pressing most of the colour compounds remain in the pomace, which is why they constitute a valuable secondary raw material and are used, among others, for producing dark anthocyanin dyes [11,12].

With the development of a healthy and balanced diet as well as consumers’ better understanding of the relationship between diet, health and the environment, innovative, healthy and environmentally friendly food products are being developed more and more, which perfectly fits into the sustainable development policy. New sources of dietary fibre, such as fruit and vegetable by-products, may be added to food products as cheap and low-calorie substances to partially replace flour, fat or sugar [13]. Most fruit and vegetable by-products account for high nutritional value, which is useful when designing new products. One option is to dry them and eat as snacks. On the other hand, a more unconventional solution is to obtain dried gels or foams which, after freeze-drying, may become one of the meals or supplement a meal by taking the form of a snack [14]. The addition of pomace also enhances the colour of food products, making them look more attractive and tastier, e.g., juice or wine is less tart or has a deeper and more distinct flavour than the counterparts without pomace. The pomace also improves the rheological properties (for example it is the case with bakery products or snacks) and may also replace ingredients harmful to people with celiac disease or gluten intolerance. Supplementing food products available in the market with natural sources of nutrients is a way to minimise the content of artificial ingredients and improve the health-promoting properties of those products. The pomace is also used in medicine, in cancer therapy or in the production of antidiabetic drugs [15,16]. The growing demand for healthy snacks combined with product innovation is a major driver to increase consumer interest in dried fruit. Sustainable and ethical production is also becoming an important aspect for European traders and consumers. Freeze-dried food, that is characterised by high quality, easy storage, universal application and a long shelf life, is an example of innovative food. The dry product is usually very porous, brittle, hygroscopic, and has excellent rehydration capacity. Freeze-dried food products may be produced by means of freeze-drying gels consisting of fruit concentrate or puree. Those products are dry and crunchy, with a high content of nutrients [17].

The aim of the work has been to develop a product design and technology for the production of food products in the form of bars, with the use of by-products derived from fruit processing. The bars were preserved in the freeze-drying process and tested for selected physical and organoleptic properties.

## 2. Results and Discussion

### 2.1. The Dry Matter Content and Water Activity of Freeze-Dried Bars with Fruit Pomace

In Table 1 samples symbol and formulations were presented.

The water content in food products is one of the main criteria determining their nutritional value, quality and storage stability. Gronowska-Senger et al. [18] explain that the higher the percentage of water in a product, the greater the risk of microbial growth. It has been shown that sodium alginate increase has no effect on water content of samples before freeze-drying whereas apple or chokeberry pomace addition increase in most cases cause significant decrease in water content (Table 2). For freeze-dried fruit snacks with apple or chokeberry pomace the dry matter content is high. Given the samples with the same apple pomace content, the increase in sodium alginate addition from 1.5 to 2% in most cases causes the dry matter content to reduce statistically significantly. Taking into account the effect of the amount of added apple pomace on the dry matter content of bars with the same sodium alginate content, in most cases differences are statistically insignificant. In the case of bars with the addition of chokeberry pomace it has been shown that the increase in the amount of added hydrocolloid and pomace causes the statistically significant increase in the dry matter content in the case freeze-dried products.

The sample with the addition of 1% sodium alginate and 2% pomace (apple or chokeberry) (1_2) proves high water content before freeze-drying but the highest dry matter content after freeze-drying may be connected with the structure of the samples that are characterised by high porosity and fragile walls separating empty spaces (Figure 1). Cell walls are much thinner; thus, water is removed easily. Marciniak-Łukasiak [19] investigated the water content in cereal bars with the constant concentration of apple and chokeberry pomace, while the proportions of instant flakes to expanded seeds were changed. With the increase in the percentage share of processed seeds, the dry matter content increased, ranging from 94% to 96%, which is consistent with the results for the freeze-dried bars under consideration, where the dry matter content increases with the increase in the pomace addition. Karwacka et al. [20] obtained similar dry matter content values for freeze-dried vegetable bars with 2% apple pomace. Ciurzyńska et al. [21] obtained similar dry matter content for freeze-dried vegetable snacks with 1.5% sodium alginate (96.72–97.67%). In dried products, and particularly in freeze-dried ones, the dry matter content is at the level of 95–99.5% [22]. Therefore, it is plausible to confirm that the freeze-drying is effective in the area of water removal from the product and ensures long-term shelf life.

The state of water in food changes continuously from the most orderly, with low water content, to that of pure water. The description of this state is expressed in terms of water activity [23], which is an important parameter for the stability of properly prepared dried fruit. Its low value guarantees the inhibition of the course of chemical and enzymatic reactions. Microbiological safety is considered to be achieved when the water activity is below 0.6. Under such conditions, microbes cannot grow. This parameter also has a significant impact on the sensory evaluation, and in particular largely contributes to shaping the sensory perception of some texture features [24,25,26]. Water activity of freeze-dried fruit snacks ranges between 0.099–0.148 for samples with the addition of apple pomace and 0.106–0.159 for samples with the addition of chokeberry pomace (Table 2). For snacks with apple pomace it has been shown that the increase in pomace addition significantly causes the freeze-dried bars water activity to decrease, while the increase in hydrocolloid centration in most cases has not caused any significant changes in the tested parameter. In the case of bars obtained with the addition of chokeberry pomace, it has been shown that increasing the addition of pomace by 3–9% and sodium alginate from 1.5% to 2% statistically significantly reduces the water activity. Marciniak-Łukasiak et al. [19] studied cereal bars with the addition of apple and chokeberry pomace. For all cereal bars, the water activity value decreased with the increase in the percentage share of expanded seeds. The water activity in which most microbes were able to grow ranged from 0.990 to 0.995. All the tested samples had the water activity below 0.6, which ensured their microbio-logical safety and may have indicated their stability during storage, provided that a sealed package was used, protecting against air penetration. Similar conclusions were also drawn by Karwacka et al. [27] for freeze-dried vegetable bars with apple pomace. On the basis of the obtained results of water activity, it is plausible to state that the tested freeze-dried bars are not susceptible to any colour change because the water activity at the value below 0.2 does not cause any non-enzymatic browning reactions to occur.

### 2.2. The Structure of Freeze-Dried Bars with Fruit Pomace

Karwacka et al. [27] indicate that the texture of food is an important factor influencing its attractiveness to consumers. It has been shown that the structure of freeze-dried fruit bars is porous (Figure 1). With the increase in pomace concentration the structure is more irregular and compact. In the case of samples with chokeberry pomace, the structure is more closed than the samples with apple pomace. The structure of samples with the lowest value of sodium alginate and pomace 1_2 is the most delicate. As far as freeze-dried vegetable bars with apple pomace are concerned, Karwacka et al. [27], also observed the heterogeneous structure of the samples and indicated that the method of their preparation (mixing or grinding) resulted in the fact that the pores were disordered and irregular. They found that the internal structure of the obtained freeze-dried bars differed from the microstructure of freeze-dried plant tissue, where, after sublimation of water, cell walls built a porous skeleton of dried products. Comparing the internal structure of vegetable bars with pomace without the of hydrocolloids [27] and fruit bars with apple or chokeberry pomace with the addition of sodium alginate as a skeleton-forming component, it was found that the addition of sodium alginate allowed to obtain blended products characterised by a porous structure similar to the structure of plant tissue, which was observed in the previous studies conducted by Ciurzyńska et al. [21,28]. The presence of the hydrocolloid also protects the product structure against breakdown during freeze-drying process by increasing the phase transition temperature [27,29].

### 2.3. The Colour of Freeze-Dried Bars with Fruit Pomace

In the case of fruit bars with the addition of apple pomace, the lightness coefficient [L*] ranges from 70.4–61.3 units, the percentage share of red colour coefficient [a*] 7.8–3.8 units, and the yellow colour coefficient [b*] 28.8–26.2 units (Figure 2). The highest value of the parameter [L*], that is responsible for the brightness, is ensured by samples 2_3A, and 1_2A, while the lowest by sample 1.5_9A. Increasing the addition of apple pomace significantly decreases the value of the [L*] parameter, thus the samples are darker. In the case of varied addition of hydrocolloid, the samples with its 2% addition are brighter than those with 1.5% addition. The highest value of the red colour coefficient [a*] is proven by sample 1.5_9A, while the lowest by sample 2_3A. Increasing the addition of pomace and hydrocolloid in most cases results in the increase in the colour index [a*] and no significant changes in the colour parameter [b*].

In the case of fruit bars with the addition of chokeberry pomace, the lightness coefficient [L*] ranges from 43.7–28.9 units, the percentage share of the red colour [a*] 20.6–14.4 units, and the yellow colour coefficient [b*] 8.6–5.3 units (Figure 3). The highest value of the parameter [L*] is characteristic for sample 1_2C, while the lowest for sample 1.5_9C. Increasing the addition of chokeberry pomace significantly decreases the lightness factor [L*]. It has been found that the samples with 2% sodium alginate are brighter than those with 1.5% additive. The highest value of the red [a*] and yellow [b*] colour index is obtained for sample 1.5_3C and 1_2C. Increasing the addition of chokeberry pomace significantly decreases the value of [a*] and [b*] coefficient, while the increase in the addition of hydrocolloid in most cases does not cause any significant changes in the index [a*] but decreases [b*] coefficient. The lowest value of the colour parameter [a*] and [b*] is proven by the samples with 9% addition of pomace.

With the amount of water removed, some raw materials become lighter and others darken depending on the form in which they are present. The decrease in the values of the yellow colour coefficient [b*] and the red colour coefficient [a*] may indicate the degradation of some naturally occurring dyes, while their increase indicates the occurrence of non-enzymatic browning reactions [30]. Ciurzyńska and Lenart [31] investigated the freeze-dried strawberry gels, and showed that the aeration time and the type of hydrocolloid significantly affected their brightness [L*], so the values of respective colour parameters were influenced by the product design, sample form, the degree of aeration and the method of drying. Ciurzyńska and Lenart [32], examining the colour of freeze-dried strawberry jellies, found that the addition of chokeberry juice concentrate significantly decreased the brightness factor of strawberry jelly by about 50 units. Increasing the percentage share of chokeberry juice concentrate from 5% to 10% resulted in obtaining a brightness factor lower by 2 units. The red colour index was 26 units, and the addition of chokeberry juice concentrate also reduced the value of this index. It was similar in the case of freeze-dried fruit bars with the addition of pomace, the brightness of which decreased with the addition of pomace. The highest values of lightness coefficient for samples with the lowest concentration of hydrocolloid and pomace (1_2) may have been connected with the structure of those samples, that was more delicate and reflected light to a varied degree as compared to the samples with higher concentration of solid ingredients.

To evaluate the effect of the addition of pomace and hydrocolloid on the colour change of the surface of fruit bars, the colour saturation index has been calculated (Table 3). In the case of the variants with the addition of apple pomace, the lowest saturation index (SI) value is proven for sample 1_2A, which is statistically significantly different from the rest. Increasing the addition of apple pomace and hydrocolloid in most cases does not cause significant changes in the examined parameter. In the case of samples with the addition of chokeberry pomace, the highest SI value is proven for variants 1.5_3C and 1_2C, while the lowest for variants with 9% of pomace. Increase in the addition of chokeberry pomace significantly decreases the colour saturation index, while the increase in the hydrocolloid concentration in most cases does not cause significant changes in this parameter.

### 2.4. The Mechanical Properties of Freeze-Dried Bars with Fruit Pomace

The mechanical properties of the fruit bars have been tested by means of a cutting test. Based on the obtained data, the maximum cutting force has been determined. The test means cutting the bar until it is cut to a depth of 10 mm, in 10 replications for each variant to serve the purpose of drawing representative curves. During the drying process, water loss and material shrinkage occur, which affects the mechanical properties and changes in tissue hardness [31]. The course of the cutting curves is similar for freeze-dried fruit bars with the addition of apple (Figure 4) and chokeberry pomace (Figure 5). With the increase in pomace concentration, the hardness of samples increases, and the higher cutting force value is necessary to destroy samples. It has been found that both the cutting curves of the bars with the addition of apple pomace and the bars with the addition of chokeberry pomace are characteristic of an irregular course. At the beginning of the process, the increase in the cutting force is observed, which is related to overcoming the resistance of the surface layers (Figure 4 and Figure 5). The applied blade destroys the cellular structures. The degree of damage depends on the structure and strength of the tissue. For all variants, after reaching the maximum force, a period of constant force is shown. This means that the structure is more homogeneous under the surface layer. Differences in force values result from diverse textures of individual variants. In the case of variants with the addition of apple pomace, the curves are flat, while samples with the addition of chokeberry pomace are characterised by a dynamic course of curves, which indicates a compact structure and higher hardness of the samples (Figure 5). Variants 2_9A (Figure 4) and 2_9C (Figure 5) are characterised by the highest hardness. Konrade et al. [33] also showed that the increase in apple pomace flour from 5 to 15% addition of pomace in cereal crispbreads increased the hardness of products. Only freeze-dried bars with the lowest concentration of hydrocolloid and pomace (1_2A and 1_2C) were characterised by the most delicate structure and required the lowest maximum cutting force, which indicated thinner walls separating the pores and easier destruction of the structure under the influence of the applied force.

The maximum cutting force of freeze-dried fruit snacks is 30–82 N for bars with apple pomace (Figure 6) and 23–68 N (Figure 7) for bars with chokeberry pomace. In both cases, sample 1_2 proves the lowest shear force, while the sample with 9% pomace is characterised by the highest cutting force. The amount of added hydrocolloid in most cases is statistically insignificant, while the amount of pomace is significant, and the larger the amount of pomace is, the stronger the cutting force is. Karwacka et al. [20] showed that freeze-dried vegetable bars with apple pomace obtained lower hardness than samples without pomace. Ciurzyńska et al. [21], using a deformation test to investigate the mechanical properties of vegetable snacks based on sodium alginate, obtained the maximum cutting force in the range from 13.16 to 13.29 N, which accounted for lower results than those obtained for fruit bars with the addition of fruit pomace. The differences in the results may be related to structure. Pomace addition probably strengthened the structure that had more resistance to cutting. Karwacka et al. [20] indicated that the addition of fruit pomace as a structure and texture forming carrier agent in dairy, meat, bakery and pastry products raised much interest among researchers whereas there was no research available on using pomace as a carrier agent in freeze-dried products.

### 2.5. The Organoleptic Evaluation of Freeze-Dried Bars with Fruit Pomace

The 6 variants of fruit snacks that have proven to be the best ones have been subjected to organoleptic evaluation, namely 3 of them with the addition of apple pomace and the other 3 with the addition of chokeberry pomace, the composition of which is presented in Table 1. Such features as general impression, crispness, taste, aroma and colour have been distinguished. A five-point rating scale has been applied, with 5 being the best one and 1 representing the worst one [34]. The results of the organoleptic evaluation of respective features of the snacks with the addition of apple and chokeberry pomace are high (Figure 8). The colour of the snacks depends on the addition of pomace because the samples containing 2% addition of pomace characterised by higher brightness (L*) have ensured more effective evaluation. Due to the composition of the bars and the freeze-drying process, their aroma is mild and may have become less perceptible, which may explain the scores that do not exceed the value of 3 points. The taste is very highly rated, especially in the case of the samples with 2% addition of pomace. The samples with the 3% of pomace are more compact, less fragile and more durable in the cutting test than those with the 2% additive, that have a softer structure, which contributes to better crispness scores. Taking into account the general impression, the best rated variants are those with the addition of 2% pomace.

The statistical analysis shows no differences between the ratings of bars with 3% addition of pomace. The variant with 2% addition of pomace, both in the case of bars with apple or chokeberry pomace, differs statistically significant from the others and proves to be the best one (Table 4).

Silva-Espinoza et al. [35] indicated that the attractiveness of dried snacks was connected with their textural and sensory properties. Consumers are interested in products that are the source of pleasure when being eaten, have rich flavour and aroma, crispness whereas many of them are also interested in the nutritional values of food. Parra et al. [36] showed that sugar snap cookies with apple pomace raised much interest among consumers, which indicated that samples with apple pomace obtained higher sensory assessment than the samples without it. In general, bars with fruit pomace are of interest to potential customers. The obtained results of organoleptic evaluation were influenced by the addition of pomace, external appearance, crispness, as well as individual preferences of the evaluators. There was no correlation between gender and education and the way the samples were assessed. The pomace concentration increases decreased crispness and general impression, which was connected with harder structure of samples in comparison to bars with the lowest hydrocolloid and pomace addition. Summarising the above results in terms of physical and organoleptic properties, the best option was the one with the addition of 1% sodium alginate and 2% pomace (1_2), both in the case of adding apple and chokeberry pomace. It was characterised by the best product design that was the most optimal due to the properties of the bar.

## 3. Materials and Methods

### 3.1. Material

#### Characteristics of Ingredients

The raw material used for the research purposes included fruit bars obtained with the use of the following ingredients:Golden Delicious apples from one production batch purchased in November 2020 constituted the basic raw material for obtaining freeze-dried food products. The fruits had been stored at the temperature of about 4 °C until the tests were carried out. Apple pulp served as the basic ingredient of the bars. Some of the apples were used for obtaining apple juice.In order to enrich the taste of snacks and to use by-products from fruit and vegetable processing, dried chokeberry or apple pomace were used after having been purchased from Greenherb Company (Wysoka, Poland) in 2021.Sodium alginate (Agnex, Białystok, Poland) and calcium lactate (Hortimex, Konin, Poland) were used as binders to initiate gelation.

### 3.2. Methodology

#### 3.2.1. Technological Methods

##### Preparation of Bars

The first step in the technological process of obtaining fruit snacks in the form of bars was to peel the apples, remove the seed chamber and chop them. Next the apples were ground into a smooth mass in a Thermomix TM31-1 made by Vorwek (Wuppertal, Germany) at the temperature of 70 °C, at the average speed of the stirrer, for about 15 min. In each of the variants, apple pulp constituted 50% of all ingredients. Some of the apples had their cores removed and chopped, and then pressed in a GOTIEGSJ-620 Evergreen (Gliwice, Poland) slow juicer to obtain juice. The rest of the ingredients included water, hydrocolloid (sodium alginate), calcium lactate and pomace (apple or chokeberry) (Table 1). The next step was to heat the apple pulp, juice and water up to the temperature of 70 °C. After reaching the appropriate temperature, juice, water, pomace and sodium alginate were successively added to the pulp. The level of the addition of the texturising substance was determined experimentally. The goal was to obtain a compact gel that would retain its shape once removed from the mould (Figure 9) [21].

The percentage share of sodium alginate was 2%, 1.5% and 1%. The whole of it was subsequently blended by means of a BOSH MaxoMixx 750 W blender for about 2 min. After that, 0.1% addition of calcium lactate was applied after having been previously dissolved in a small amount of water before being added to the other ingredients. It was then blended for about 2 min.

##### Freeze-Drying

The obtained gels were poured into silicone moulds of 14 × 10 × 2.5 cm in dimensions, manufactured by Tescoma (Poland). The semi-finished product was cooled down to reach the ambient temperature, removed from the moulds to be subsequently frozen in an Irinox freezer (Treviso, Italy) at −40 °C for 2 h.

After freezing, the fruit bars were placed on the shelves of a Christ ALPHA 1–4 freeze dryer (Osterode am Harz, Germany). In the course of drying, the pressure in the freeze dryer chamber was 63 Pa, and the temperature of the heating plates of the freeze dryer was 30 °C. The drying process lasted 48 h, while the pressure was 63 Pa. After freeze-drying, the bars kept their shape of a rectangle showing the dimensions of 14 × 10 × 2.5 cm. The photos of the bars are included in Table 1.

The technological scheme of obtaining freeze-dried food products is shown in Figure 10, while the designs of respective snack variants are presented in Table 1.

#### 3.2.2. Analytical Methods

##### Determination of Physical Properties of Obtained Snacks

Water activity was determined by means of a Rotronic HygroLab C1 apparatus (Bassersdorf, Switzerland) in accordance with the manufacturer’s instructions. The snacks were chopped and placed in a dish that was an integral part of the device. Samples of approximately ¾ of the volume occupied the cell. Three repetitions of the measurement were performed for each variant. The measurement time was approximately 4–5 min.

Dry matter content. The dry substance content was measured by means of the drying method. About 1 g of bars—before and after freeze-drying—were weighed into weighing vessels of pre-set weight, and then placed in a convection dryer WAMED SUP 65 W/G (Warsaw, Poland) to be treated at the temperature of 70 °C. The determination process took 24 h, after which the sample vessels were placed in a desiccator to cool down, and subsequently reweighed to calculate the dry matter content [37]. Measurement was repeated three times for each product design.
(1)$$s.s=\left[\frac{\left(w_{3}-w_{1}\right)}{\left(w_{2}-w_{1}\right)}\right]\times 100$$

The water content [%] for samples before freeze-drying was calculated on the basis of equation
(2)$$H_{2}O=100-s.s.$$
where:s.s—dry matter content [%],w_1_—the mass of the empty vessel [g],w_2_—weight of the sample dish before drying [g],w_3_—weight of the sample dish after drying [g].

Microscopy. The changes in structure in terms of the effect of the composition were determined by means of a scanning electron microscope TM-3000 HITACHI (Fichtenhain, Germany) [34]. Out of the freeze-dried gel cubes, a piece of about 1–2 mm was cut out across the cube. Structural changes were determined at 100× magnification.

Colour. The colour change was assessed using a Chroma-Meter CR-300 made by Minolta (Tokyo, Japan). The material was tested in the CIE L*, a*, b* system with the following parameters: observer 2°, measuring diameter 8 mm and the light source D65 [38]. In the CIE L*, a*, b* system, the L* parameter means the brightness of the sample, the higher the value of the L* parameter, the closer the colour of the sample to white (range 0–100 units). The a* and b* coordinates, on the other hand, take positive and negative values. The a* coordinate indicates the proportion of red in the case of positive values and the proportion of green in the case of negative values. Correspondingly, the b* parameter relates to the percentage share of yellow and blue colours. Ten measurements were made in various places on the surface of the obtained bars. Based on the L*, a*, b* parameters, the colour saturation (SI) was calculated.

SI—color saturation indicator [39]
(3)$$\mathrm{SI}=\sqrt{{\left(a^{*}\right)}^{2}+{\left(b^{*}\right)}^{2}}$$
where:a*—red colour coefficient [dimensionless value]b*—yellow colour coefficient [dimensionless value]

Mechanical properties. The study of mechanical properties was checked using a TA.HD plus Texture Analyzer (Stable Micro Systems, Surrey, UK) at an ambient temperature. The cutting test was performed using a knife 62 mm long, 24 mm wide and 0.5 mm thick. The test was carried out until the bar was cut to the depth of 10 mm, with a head travel speed of 1.0 mm/s. The test results, that is strength, distance and time, were recorded by the computer software program Texture Export [31]. The determination was performed in 10 replications for each variant. Based on the obtained data, the maximum drought cutting force was determined and the curves of mechanical properties were drawn.

Organoleptic evaluation. The organoleptic evaluation of the obtained food products was carried out using a 5-point rating scale, where 1 meant the least favourable impression, and 5 the most favourable one [36]. The following features were distinguished: general appearance, colour, taste, smell, crispness and general impression [40]. The group of evaluators consisted of 10 students from the Warsaw University of Life Sciences (SGGW). Six samples, that had been appropriately coded, were subjected to evaluation.

##### Statistical Analysis

The results were statistically analysed by means of the Statistica Statgraphics XVII software program. For this purpose, the one-way analysis of variance (ANOVA) was used and homogeneous groups were determined using the F test.

## 4. Conclusions

The increase of apple or chokeberry pomace concentration decreased the water activity and brightness, whereas in most cases the dry matter content of freeze-dried bars increased. Mechanical properties vary depending on the product design. With the increase in the amount of pomace, the shear force increases, which is influenced by the structure of the samples. The variant with the 2% addition of pomace is characterised by the most delicate structure that influences mechanical properties. Organoleptic evaluations confirm the high quality of freeze-dried bars with the addition of fruit pomace. Taking into account the results of determinations and organoleptic evaluation, the variant with 1% addition of sodium alginate and 2% addition of pomace has proven to be the best variant in terms of physical and organoleptic properties.

## Figures and Tables

**Figure 1 molecules-27-03095-f001:**
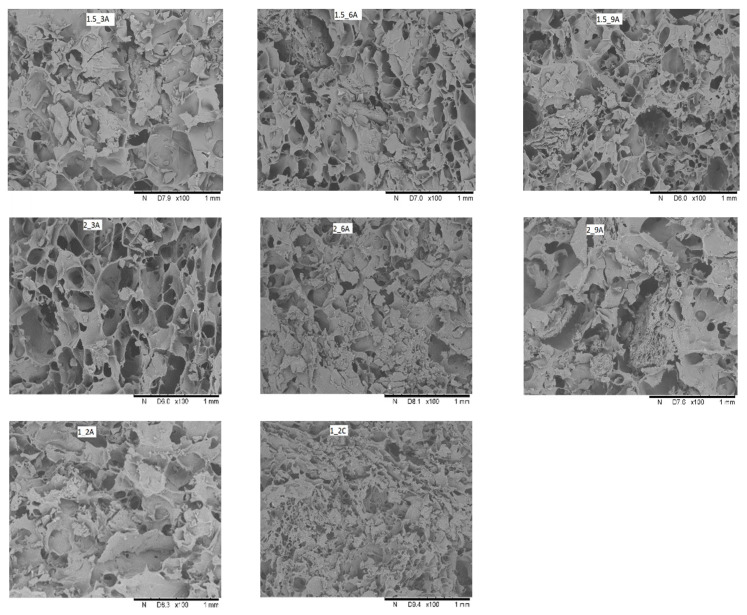
The effect of the amount of hydrocolloid and pomace addition on the structure of freeze-dried fruit bars with apple and chokeberry pomace at 100× Magnification. Denotations in Table 1 include Index (A) next to the sample symbol—bars with the apple pomace, Index (C) next to the sample symbol—bars with the chokeberry pomace.

**Figure 2 molecules-27-03095-f002:**
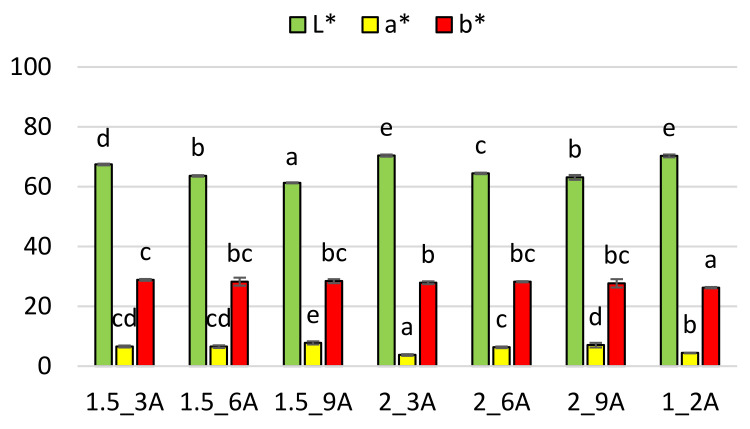
The effect of the amount of hydrocolloid and pomace addition on the colour in freeze-dried fruit bars with apple pomace. Mean values marked with the same index letter (a–e) do not differ statistically significantly at the level of *p* = 0.05. Denotations in Table 1 include Index (A) next to the sample symbol—bars with the apple pomace.

**Figure 3 molecules-27-03095-f003:**
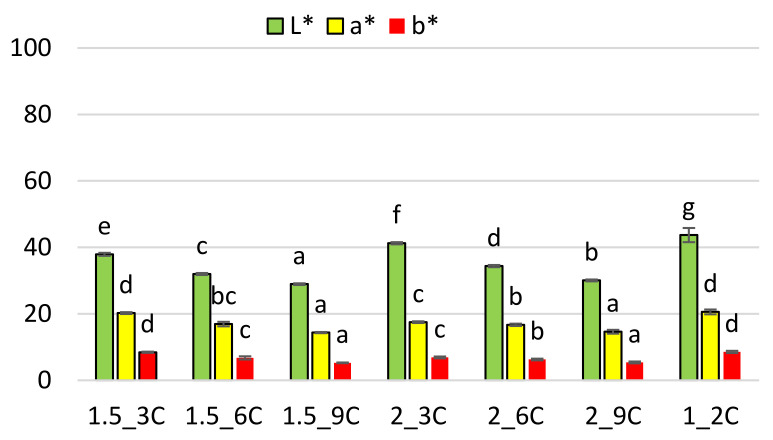
The effect of the amount of hydrocolloid and pomace addition on the colour in freeze-dried fruit bars with the addition of chokeberry pomace. Mean values marked with the same index letter (a–g) do not differ statistically significantly at the level of *p* = 0.05. Denotations in Table 1 include Index (C) next to the sample symbol—bars with the chokeberry pomace.

**Figure 4 molecules-27-03095-f004:**
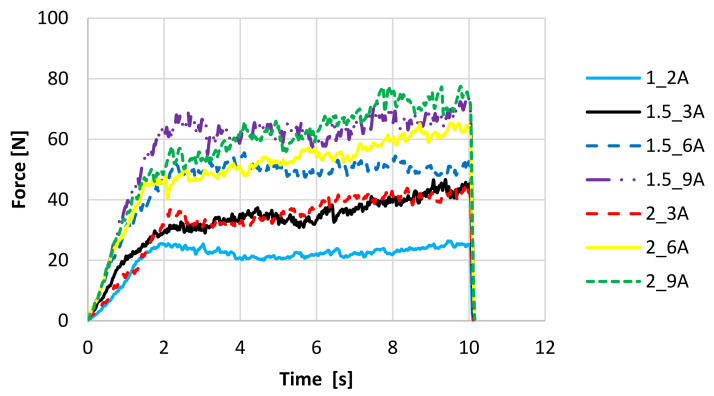
The cutting curves of freeze-dried fruit bars with apple pomace. Denotations in Table 1 include Index (A) next to the sample symbol—bars with the apple pomace.

**Figure 5 molecules-27-03095-f005:**
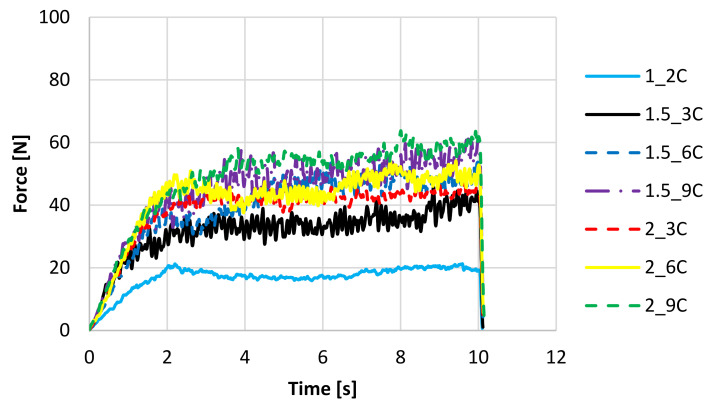
The cutting curves of freeze-dried fruit bars with chokeberry pomace. Denotations in Table 1 include Index (C) next to the sample symbol—bars with the chokeberry pomace.

**Figure 6 molecules-27-03095-f006:**
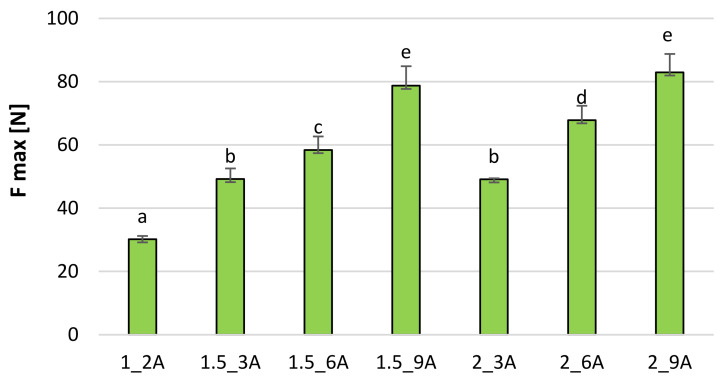
The maximum cutting force [N] of freeze-dried fruit bars with apple pomace. Mean values marked with the same index letter (a–e) do not differ statistically significantly at the level of *p* = 0.05. Denotations in Table 1 include Index (A) next to the sample symbol—bars with the apple pomace.

**Figure 7 molecules-27-03095-f007:**
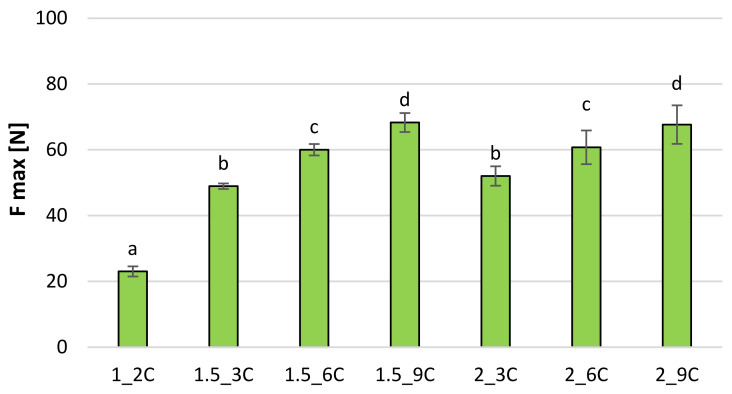
The maximum cutting force [N] of freeze-dried fruit bars with chokeberry pomace. Mean values marked with the same index letter (a–d) do not differ statistically significantly at the level of *p* = 0.05. Denotations in Table 1 include Index (C) next to the sample symbol—bars with the chokeberry pomace.

**Figure 8 molecules-27-03095-f008:**
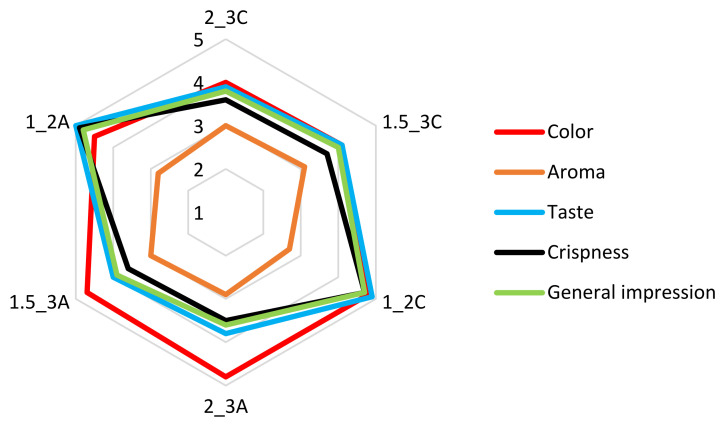
The average results for the organoleptic discriminants of fruit snack. Denotations in Table 1 include Index (A) next to the sample symbol—bars with the apple pomace, Index (C) next to the sample symbol—bars with the chokeberry pomace.

**Figure 9 molecules-27-03095-f009:**
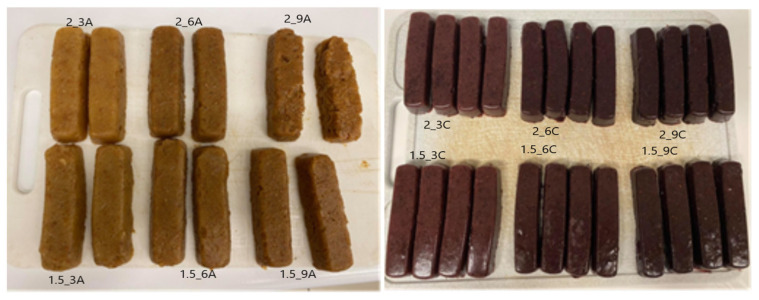
Photos of bars with apple pomace (A) and chokeberry pomace (C) after gelation and after freeze-drying. Denotations in Table 1 include Index (A) next to the sample symbol—bars with the apple pomace, Index (C) next to the sample symbol—bars with the chokeberry pomace.

**Figure 10 molecules-27-03095-f010:**
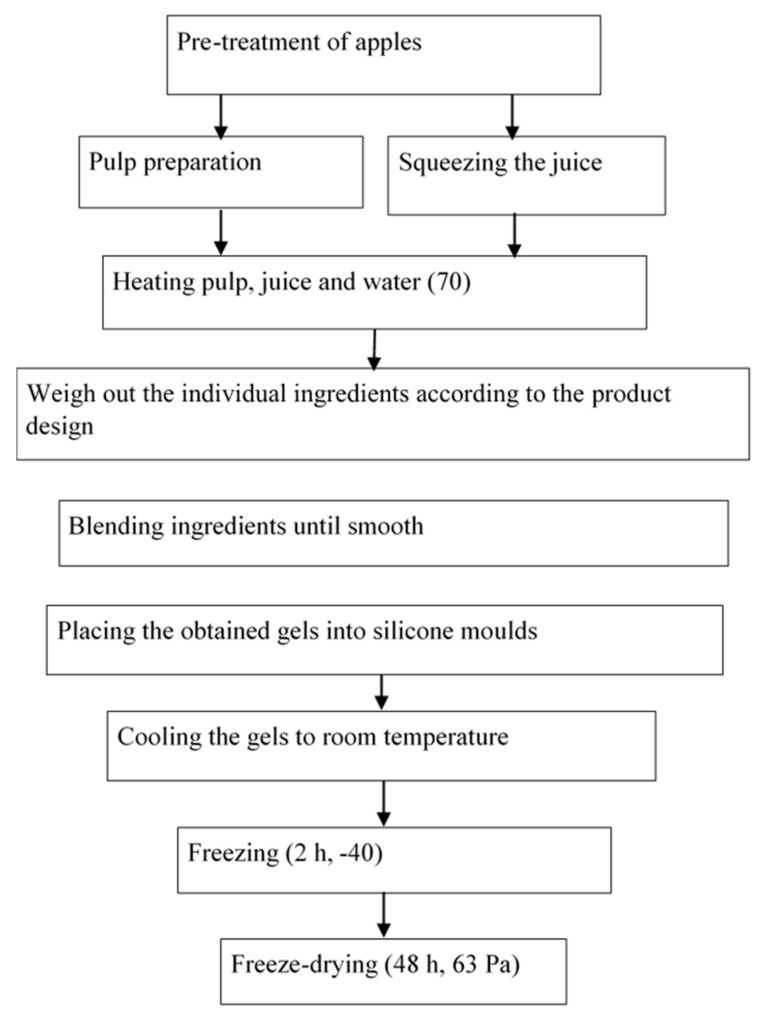
Process diagram for obtaining freeze-dried food products.

**Table 1 molecules-27-03095-t001:** Formulation of individual bar variants. Index (A) next to the sample symbol—bars with the apple pomace, Index (C) next to the sample symbol—bars with the chokeberry pomace.

Sample Symbol	Apple Pomace	Chokeberry Pomace	Apple Pulp	Apple Juice	Water	Calcium Lactate	Sodium Alginate
2_3A	3%	-	50%	22.5%	22.4%	0.1%	2%
2_3C	-	3%					
2_6A	6%	-	50%	21%	20.9%	0.1%	2%
2_6C	-	6%					
2_9A	9%	-	50%	19.5%	19.4%	0.1%	2%
2_9C	-	9%					
1.5_3A	3%	-	50%	22.7%	22.7%	0.1%	2%
1.5_3C	-	3%					
1.5_6A	6%	-	50%	1.2%	21.2%	0.1%	2%
1.5_6C	-	6%					
1.5_9A	9%	-	50%	19.7%	19.7%	0.1%	2%
1.5_9C	-	9%					
1_2A	2%	-	50%	23.6%	23.4%	0.1%	2%
1_2C	-	2%					

**Table 2 molecules-27-03095-t002:** The effect of hydrocolloid and pomace addition amount on the dry matter content and water activity of freeze-dried fruit bars with apple or chokeberry pomace and water content of samples before freeze-drying. Denotations in Table 1 include. Index (A) next to the sample symbol—bars with the apple pomace, Index (C) next to the sample symbol—bars with the chokeberry pomace. Mean values marked with the same index letter (a–e, A–F) do not differ statistically significantly at the level of *p* = 0.05. The small index letter (a–e) indicates the statistical analysis results for samples with apple pomace whereas the big index letter (A–F) indicates the statistical analysis results for samples with chokeberry pomace.

	Freeze-Dried Samples		Samples before Freeze-Drying
Sample	Average Dry Matter Content	Average Water Activity	Average Water Content
1.5_3A	98.23% ± 0.013 ^a,b^	0.14 ± 0.002 ^d^	77.12% ± 0.086 ^c,d^
1.5_6A	98.40% ± 0.055 ^b,c^	0.12 ± 0.001 ^b^	74.74% ± 0.036 ^b,c^
1.5_9A	98.56% ± 0.013 ^c^	0.099 ± 0.001 ^a^	71.92% ± 0.006 ^a^
2_3A	97.99% ± 0.009 ^a^	0.148 ± 0.003 ^e^	77.26% ± 0.057 ^c,d^
2_6A	98.07% ± 0.01 ^a^	0.126 ± 0.002 ^b^	75.00% ± 0.03 ^a,b^
2_9A	98.17% ± 0.087 ^a,b^	0.102 ± 0.003 ^a^	71.31% ± 0.314 ^a^
1_2A	99.46% ± 0.28 ^d^	0.132 ± 0.001 ^c^	81.07% ± 1.078 ^d^
1.5_3C	97.11% ± 0.00 ^A^	0.15 ± 0.001 ^E^	79.572% ± 0.411 ^C^
1.5_6C	97.90% ± 0.014 ^C^	0.116 ± 0.002 ^C^	73.178% ± 0.317 ^B^
1.5_9C	98.04% ± 0.034 ^D^	0.106 ± 0.001 ^A^	70.98% ± 0.192 ^A^
2_3C	97.28% ± 0.038 ^B^	0.159 ± 0.001 ^F^	76.90% ± 0.063 ^C^
2_6C	98.00% ± 0.022 ^D^	0.126 ± 0.000 ^D^	71.65% ± 0.526 ^B^
2_9C	98.11% ± 0.046 ^E^	0.112 ± 0.001 ^B^	70.32% ± 1.298 ^A^
1_2C	98.29% ± 0.000 ^F^	0.116 ± 0.001 ^B,C^	78.087% ± 0.376 ^C^

**Table 3 molecules-27-03095-t003:** Colour saturation index [SI] of freeze-dried fruit bars. Mean values marked with the same index letter (a–d) do not differ statistically significantly at the level of *p* = 0.05. Denotations in Table 1 include Index (A) next to the sample symbol—bars with the apple pomace, Index (C) next to the sample symbol—bars with the chokeberry pomace.

Sample Symbol	Samples with Chokeberry Pomace [C]	Samples with Apple Pomace [A]
1.5_3	21.9 ± 0.33 ^d^	29.6 ± 0.37 ^c^
1.5_6	18.2 ± 0.78 ^b,c^	29.0 ± 1.38 ^b,c^
1.5_9	15.3 ± 0.18 ^a^	29.5 ± 0.77 ^c^
2_3	18.8 ± 0.35 ^c^	28.2 ± 0.49 ^b^
2_6	17.8 ± 0.43 ^b^	28.9 ± 0.26 ^b,c^
2_9	15.6 ± 0.63 ^a^	28.6 ± 1.54 ^b,c^
1_2	22.3 ± 0.81 ^d^	26.6 ± 0.24 ^a^

**Table 4 molecules-27-03095-t004:** The average results of organoleptic evaluation for freeze-dried bars. Denotations in Table 1 include Index (A) next to the sample symbol—bars with the apple pomace, Index (C) next to the sample symbol—bars with the chokeberry pomace. Mean values marked with the same index letter (a,b) do not differ statistically significantly at the level of *p* = 0.05. The small index letter (a,b) indicates the statistical analysis results for samples with apple pomace whereas the big index letter (A,B) indicates the statistical analysis results for samples with chokeberry pomace.

	Sample Symbol					
	1_2A	1.5_3A	2_3A	1_2C	1.5_3C	2_3C
Color	$4.5\;\pm \;0.5\;{}^{a}$	$4.7\;\pm \;0.5\;{}^{a}$	$4.8\;\pm \;0.4\;{}^{a}$	$4.8\;\pm \;0.4\;{}^{B}$	$4.1\;\pm \;0.3\;{}^{A}$	$4\;\pm \;0.7\;{}^{A}$
Aroma	$2.8\;\pm \;0.4\;{}^{a}$	$3\;\pm \;0.7\;{}^{a}$	$2.9\;\pm \;0.7\;{}^{a}$	$2.7\;\pm \;0.5\;{}^{A}$	$3.1\;\pm \;0.7\;{}^{A}$	$3\;\pm \;0.7\;{}^{A}$
Taste	$5\;\pm \;0.0\;{}^{b}$	$4\;\pm \;0.5\;{}^{a}$	$3.8\;\pm \;0.6\;{}^{a}$	$4.9\;\pm \;0.3\;{}^{B}$	$4.1\;\pm \;0.3\;{}^{A}$	$3.9\;\pm \;0.7\;{}^{A}$
Crisspness	$4.9\;\pm \;0.3\;{}^{b}$	$3.6\;\pm \;0.5\;{}^{a}$	$3.5\;\pm \;0.5\;{}^{a}$	$4.7\;\pm \;0.7\;{}^{B}$	$3.7\;\pm \;0.5\;{}^{A}$	$3.6\;\pm \;0.7\;{}^{A}$
General impression	$4.8\;\pm \;0.4\;{}^{b}$	$3.9\;\pm \;0.6\;{}^{a}$	$3.6\;\pm \;0.7\;{}^{a}$	$4.7\;\pm \;0.5\;{}^{B}$	$4\;\pm \;0.5\;{}^{A}$	$3.8\;\pm \;0.8\;{}^{A}$
Average	4.4	3.84	3.72	4.36	3.8	3.66

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
