# Peer review of "Innovative Freeze-Dried Snacks with Sodium Alginate and Fruit Pomace (Only Apple or Only Chokeberry) Obtained within the Framework of Sustainable Production"

_molecules, 2022, doi:10.3390/molecules27103095_

Round 1

Reviewer 1 Report

The quality of the manuscript has been improved by the authors. The manuscript is interesting and valuable. In my opinion English language of the manuscript still requires corrections.

The authors should correct:

  • Table 4 "Chokeberry pomaca"
  • Figure 10 "Weigh out the individual ingredients according to the ..."

Conclusions should be corrected. In this part the authors should state the most important outcome of their work, not only summarize the points already made in the manuscript.

Author Response

We would like to express our thanks for the review. Your suggestions were analysed, all of them were taken into account in the manuscript and marked in red colour. In addition we provide here answers to specific issues.

  • In my opinion English language of the manuscript still requires corrections.

Answer: Suggestion was taken into account in manuscript. The manuscript was checked by native speaker.

The authors should correct:

  • Table 4 "Chokeberry pomaca"

Answer: Suggestion was taken into account in manuscript.

  • Figure 10 "Weigh out the individual ingredients according to the ..."

Answer: Suggestion was taken into account in manuscript.

  • Conclusions should be corrected. In this part the authors should state the most important outcome of their work, not only summarize the points already made in the manuscript.

Answer: Suggestion was taken into account in manuscript.

Reviewer 2 Report

The manuscript is very interesting, anyway there are several methodological and conceptual issues which must be taken into consideration and particularly the introduction and presentation of the results must be done in compliance with the main scope of the paper. The aim of the work has been to develop a product design and technology for the production of food products in the form of bars, with the use of by-products derived from fruit processing.

The introduction is highlighting the aspects of valuable nutrients and health-promoting ingredients, but the paper doesn’t cover these aspects and is not investigating the health promoting ingredients, antioxidant capacity of the obtained product, which means that the introduction must be modified highlighting the aspects of sustainable production and economic efficiency.

Quality control and quality assurance of analytical part is not provided.

In my opinion conclusions are very general and not specific to parameters investigated.

One of the main conclusions is that fruit snacks with the addition of pomace are characterized by favorable physical and organoleptic properties, and the obtained results are characteristic for the drying method used. Only several physical aspects are investigated in order to have such a conclusion. 

Author Response

We would like to express our thanks for the review. Your suggestions were analysed, all of them were taken into account in the manuscript and marked in red colour. In addition we provide here answers to specific issues.

  • The introduction is highlighting the aspects of valuable nutrients and health-promoting ingredients, but the paper doesn’t cover these aspects and is not investigating the health promoting ingredients, antioxidant capacity of the obtained product, which means that the introduction must be modified highlighting the aspects of sustainable production and economic efficiency.

Answer: Suggestion was taken into account in manuscript.

Quality control and quality assurance of analytical part is not provided.

  • In my opinion conclusions are very general and not specific to parameters investigated. One of the main conclusions is that fruit snacks with the addition of pomace are characterized by favorable physical and organoleptic properties, and the obtained results are characteristic for the drying method used. Only several physical aspects are investigated in order to have such a conclusion. 

Answer: Suggestion was taken into account in manuscript.

Reviewer 3 Report

THE MANUSCRIPT IS GOOD  BUT IT NEEDS REVISING AS ILLUSTRATED IN THE MANUSCRIPT  (MY COMMENTS).

Author Response

We would like to express our thanks for the review. Your suggestions were analysed, all of them were taken into account in the manuscript and marked in red colour. In addition we provide here answers to specific issues.

  • Line 10: Abstract needs data of results

Answer: Suggestion was taken into account in manuscript.

Line 14-22: “Freeze-dried bares were obtained according to sustainability production which in this case was relied on application of fruit pomace. The freeze-drying process ensures the microbiological safety of the product without the need to use chemical preservatives. Freeze-dried samples obtained low water activity in the range of 0.099-0.159. The increase in pomace concentration (3-9%) boosted the dry matter content to above 98%, and decreased the brightness of the freeze-dried bars about 6 to 10 units. Mechanical properties varied depending on the product design. With the increase in the amount of pomace, the shear force increased at 23% to 41%. Based on the results, the best variant, that has the most delicate structure and the best organoleptic properties, has proven to contain 1% sodium alginate and 2% pomace.”

  • Line 22: Put statement obout sustainability finding from this study

Answer: Suggestion was taken into account in manuscript.

Line 14-16: „Freeze-dried bares were obtained according to sustainability production which in this case was relied on application of fruit pomace.”

  • Line 26: The introduction is very long. You can reduce it.

Answer: Suggestion was taken into account in manuscript.

  • Line 102-103: When adding sodium alginate the moisture should be reduced due it has not effect on the MC?

Answer: The change in the amount of added sodium alginate was 0.5%, which could mean that in the case of samples before lyophilization, the differences in water content were not statistically significant.

  • Line 196: the results in the fig. 2 has no minus data???

Answer: There are no negative values in Figures 2 or 3 because the color index [a *] was shifted towards red.

  • Line 197: The negative data are not found in the fig. 2?

Answer: There are no negative values in Figures 2 or 3 because the color index [b *] was shifted towards yellow.

  • Line 200: Why?

Answer: The increase of pomace addition caused structure changes „With the increase in pomace concentration the structure is more irregular and compact”, and dry matter content increased, what caused darkening of samples.

  • Line 205: Fig 2, 3. Must be a star above L, a, b like L*, a*, b*

Answer: Suggestion was taken into account in manuscript.

  • Line 263: Needs SD for all data

Answer: Suggestion was taken into account in manuscript.

  • Line 319: All figs need tale care (improvement)

Answer: Suggestion was taken into account in manuscript.

  • Line 380: Why 220? Its old

Answer: Apples were bought in November 2020 and bars were produced and investigated during the master thesis which was finished on June 2021. The first manuscript submission was at 2021-10-13, so apples aren't old.

  • Line 419: Table 4. Put % here for all columns and delete from data

Answer: Suggestion was taken into account in manuscript.

  • 484: Put reference

Answer: Suggestion was taken into account in manuscript.

  • Line 508: Conclusion must be without numbering. To be paragraph

Answer: Suggestion was taken into account in manuscript.

Round 2

Reviewer 1 Report

I accept the manuscript in the present form.

Author Response

Thank you very much:)

Reviewer 2 Report

It is mentioned that suggestions were taken into consideration, but the part of quality control and quality assurance is not added. 

Author Response

In our opinion the suggestion that "quality control and quality assurance of analytical part is not provided" is to wide and imprecise. The analytical part covering the research methodology has been described in detail, giving the source of the methodology. Obtained data were analysed and compared according to data published by other researchers.